# Clinical Correlations between Serological Markers and Endometrial Cancer

**DOI:** 10.3390/cancers16101935

**Published:** 2024-05-20

**Authors:** Alina-Gabriela Marin, Alexandru George Filipescu, Răzvan Cosmin Petca, Radu Vlădăreanu, Aida Petca

**Affiliations:** 1“Carol Davila” Faculty of Medicine, Department of Obstetrics and Gynecology, University of Medicine and Pharmacy, 050474 Bucharest, Romania; alina.marin@drd.umfcd.ro (A.-G.M.); alexandru.filipescu@umfcd.ro (A.G.F.); radu.vladareanu@umfcd.ro (R.V.); aida.petca@umfcd.ro (A.P.); 2Department of Obstetrics and Gynaecology, Elias Emergency University Hospital, 011461 Bucharest, Romania; 3Department of Urology, “Prof. Dr. Th. Burghele” Clinical Hospital, 20 Panduri Street, 050659 Bucharest, Romania

**Keywords:** endometrial cancer, endometrial hyperplasia, serological markers

## Abstract

**Simple Summary:**

Endometrial cancer disrupts various internal functions, which may be reflected across several blood markers or biomarkers. The precise ratios of these biomarkers could be useful for endometrial cancer diagnosis, prognosis, and treatment decisions. Our study aimed to observe whether a panel of blood markers (red blood cells, white blood cells, platelets, inflammatory markers) managed to distinguish between two endometrial entities (neoplasia versus hyperplasia).

**Abstract:**

**Background:** Endometrial cancer is associated with changes in blood cell counts and with high levels of inflammatory markers, thus reflecting the tumor’s impact on various biological processes and suggesting their potential as biomarkers for endometrial cancer diagnosis, prognosis, and treatment response. The neutrophil-to-lymphocyte ratio, platelet-to-lymphocyte ratio, and monocyte-to-lymphocyte ratio in peripheral blood sampled preoperatively from patients have been reported to be independently associated with the prognosis of different types of malignancies. **Objectives:** This study aimed to compare several blood markers—red blood cells, white blood cells, platelet parameters, neutrophil-to-lymphocyte ratio, platelet-to-lymphocyte ratio, monocyte-to-lymphocyte ratio, C-reactive protein, and fibrinogen—in patients with benign or malignant endometrial tumors. **Material and methods:** Our retrospective study included 670 patients (192 diagnosed with endometrial cancer and 478 with endometrial hyperplasia), and we compared the serological parameters discussed above with those sampled the day before surgery. **Results:** Analysis of complete blood count indices revealed no significant differences in red blood cell or total white blood cell parameters between the endometrial cancer group and the endometrial hyperplasia group. However, a distinct pattern emerged in the white blood cell differential. The endometrial cancer group showed a statistically significant decrease in lymphocyte count compared with the endometrial hyperplasia group. In contrast, the endometrial cancer group showed significantly higher mean platelet counts and increased mean platelet volume compared with controls. Furthermore, the endometrial cancer group demonstrated a marked inflammatory response, as evidenced by significantly elevated levels of C-reactive protein, fibrinogen, neutrophil-to-lymphocyte ratio, platelet-to-lymphocyte ratio, and monocyte-to-lymphocyte ratio compared with the endometrial hyperplasia group. **Conclusions:** The current research revealed statistically significant differences in multiple serological biomarkers between the two groups. These findings support the initial hypothesis regarding the potential utility of these biomarkers in endometrial cancer diagnosis, prognosis, and treatment response, highlighting the existence of biomarkers affordable for analysis under any health system, regardless of the country’s level of development.

## 1. Background

The development and progression of endometrial cancer (EC) are intricate phenomena shaped by numerous elements that encompass genetic susceptibilities, fluctuations in hormonal levels, and exposure to diverse environmental factors.

In EC patients, outcome is related to the analysis of cancer-related risk factors such as advanced Federation of Gynaecology and Obstetrics (FIGO) stage, consisting of myometrial, cervical stromal, or lymphovascular space invasion (LVSI), extrauterine spreading of the disease, positive peritoneal cytology, positive pelvic or para-aortic ganglions, third-grade histological cells, completeness of tumoral resection, and the serum level of CA-125 [1]. The outcome can also be influenced by host-related risk factors such as serological parameters (both each component and the relationship between several elements).

Over the past few years, there has been increasing interest in the potential role of serological parameters as biomarkers for improving early EC diagnosis, refining risk stratification, prognosis, and response to treatment, especially because they provide valuable information about an individual’s overall state of health and may reflect underlying conditions, including EC [2,3,4].

EC progression is often associated with the upregulation of pro-inflammatory signaling pathways. This results in the release of a spectrum of inflammatory mediators, also known as biomarkers—C-reactive protein (CRP), interleukin-6 (IL6), and tumor necrosis factor-alpha (TNFα)—into the bloodstream. CRP, known for its sensitivity as a marker of systemic inflammation, has been thoroughly examined in relation to EC. Studies have revealed both an elevated risk of developing EC and a less favorable prognosis among diagnosed patients, highlighting a potential involvement of CRP in mediating tumor growth and proliferation [5].

A causative link between inflammation and cancer was first hypothesized in 1863 by Virchow, who observed leukocytes in neoplastic tissues [6]. Paraneoplastic leukemoid reactions, characterized by extreme leucocytosis (>40 × 10⁹/L), have been observed in up to 10% of patients with solid tumors. These reactions arise from tumor-derived cytokines, including granulocyte colony-stimulating factor (GCSF), granulocyte-macrophage colony-stimulating factor (GMCSF), interleukin-1 alpha (IL-1α), and IL6 [7]. GCSF-secreting tumors are associated with aggressive disease progression and poor prognosis, potentially due to the cytokine’s ability to stimulate tumor growth in an autocrine fashion. A review of the literature identified 22 cases of gynecological malignancies exhibiting co-occurrence of leucocytosis and elevated GCSF levels. Notably, cervical cancer was the most prevalent malignancy within this cohort (n = 14), followed by uterine cancer (n = 5) and peritoneal/ovarian cancer (n = 3) [8].

Beyond their well-known role in blood clotting and maintaining hemostasis, platelets may also play a part in EC. An elevated platelet count, or thrombocytosis [defined as greater than 400,000 platelets per microliter (μL)], has been linked to more advanced stages of EC and a poorer prognosis for the diagnosed patient [9]. Studies have further explored the association between platelet activation and endometrial abnormalities, showing that increased levels of mean platelet volume (MPV), platelet distribution width (PDW), and platelet index are significantly associated with both the presence and severity of endometrial lesions. As a routine blood test can easily obtain these parameters, they provide a potentially valuable and non-invasive tool for assessing and eventually diagnosing endometrial lesions [10,11].

Recent research has explored the utility of novel inflammatory biomarkers alongside established markers like granulocytes and platelets in EC. These emerging markers include the neutrophil-to-lymphocyte ratio (NLR, the absolute neutrophil count divided by the absolute lymphocyte count), platelet-to-lymphocyte ratio (PLR, the absolute platelet count divided by the absolute lymphocyte count), and monocyte-to-lymphocyte ratio (MLR, the absolute monocyte count divided by the absolute lymphocyte count). NLR has been shown to be an independent prognostic factor in various malignancies, including EC. Similarly, PLR has been associated with clinicopathological factors such as tumor stage, grade, and overall survival (OS) in EC patients [12]. Preoperative MLR has also emerged as an independent predictor of disease recurrence in patients with stage I endometrioid EC. This finding suggests the potential utility of MLR as a biomarker for assessing systemic inflammatory response and for risk stratification of recurrence in patients diagnosed with low-risk EC [13].

The objective of this study was to compare several blood markers [red blood cell (RBC) count, hemoglobin, hematocrit, RBC distribution width (RDW) levels, platelet count, MPV, PDW, white blood cell (WBC) count, granulocyte count, lymphocyte count, monocyte count, NLR, PLR, MLR, CRP, and fibrinogen in patients with benign and malignant endometrial pathology.

## 2. Materials and Methods

### 2.1. Study Design

A retrospective cohort study was conducted to investigate the association between serological biomarkers and the risk of EC.

### 2.2. Data Source

Anonymized electronic health records were retrieved from the database of the Obstetrics and Gynecology Clinic of Elias University Emergency Hospital, Bucharest, encompassing the period from January 2015 to December 2022.

### 2.3. Study Population

The first group included 192 women aged over 18 years old who were diagnosed with EC during the aforementioned timeframe, classified according to FIGO staging, who underwent surgery (total abdominal hysterectomy with bilateral salpingo-oophorectomy—TAH + BSO, bilateral pelvic lymph node dissection, para-aortic lymph node dissection, or peritoneal cytology examination, depending on the endometrial pathology and clinical stage).

The second group of patients included women aged over 18 years old who were diagnosed with endometrial hyperplasia (EH) during the same timeframe as above, totaling 478 women who underwent an endometrial biopsy or TAH with or without BSO (according to the age of the patient and type of EH).

Women with other gynecological cancers, incomplete records, missing data for key variables, or significant medical comorbidities were excluded from the study.

### 2.4. Data Collection

Standardized procedures were implemented to extract anonymized data from electronic medical records. The collected data included demographic information (such as age, ethnicity, education, employment, marital and living status, and the number of births), clinical and histopathological characteristics (such as body mass index (BMI), histological type, tumoral grade and stage, presence of lymph nodes or organ metastases), and treatment details, We also extracted the following hematological and biochemical markers:Complete blood count (CBC) including RBC count, hemoglobin level, hematocrit, RDW levels, WBC count, granulocyte count, lymphocyte count, monocyte count, platelet count, MPV, and PDW levels;NLR, PLR, and MLR;For immunology, CRP, and fibrinogen.

Blood specimens for the blood tests mentioned above were harvested via standard procedure from the peripheric veins of patients 24–48 h before surgical intervention. Established definitions and cut-off points were used for all markers to ensure consistency and comparability.

According to the ratios’ precise definitions, NLR was established as the absolute neutrophil count divided by the absolute lymphocyte count, PLR as the absolute platelet count divided by the absolute lymphocyte count, and MLR as the absolute monocyte count.

### 2.5. Statistical Analysis

Data analysis was performed using two established statistical software packages: XLSTAT (version 2023.3.1.1416) and Statistical Package for the Social Science (SPSS) statistical software package, version 20 (SPSS Inc., Chicago, IL, USA). Descriptive statistics were generated to characterize the study population and assess the distribution of hematological and biochemical markers. A *p*-value threshold of less than 0.05 indicated statistically significant findings.

Clinicopathological characteristics were compared between groups using appropriate statistical tests. The t-test for equality of means was employed for normally distributed continuous variables. Additionally, receiver operating characteristic (ROC) curve analysis was conducted to evaluate the diagnostic efficacy of NLR and PLR in differentiating the EC from the EH group.

Results are presented as mean ± standard deviation (SD) for normally distributed data and median (interquartile range, IQR 1 and 3). This approach comprehensively explains each group’s central tendency and data dispersion.

### 2.6. Confidentiality and Ethics

The study was granted ethical approval by the institutional review board of Elias University Emergency Hospital, Bucharest. Participants’ data were de-identified to protect their privacy. The study was conducted in accordance with the principles of the Declaration of Helsinki.

## 3. Results

In our research, we included a total of 670 patients. We divided the participants into two groups; group 1 consisted of 192 patients with EC and group 2 included 478 patients with EH. The mean age in the EC group was 62.42 years ± 10.61, compared with 53.12 years ± 10.76 for the EH groujp. Patients in the EC group reached menarche at a significantly younger age (11.11 years ± 1.10 vs. 13.77 years ± 1.12 in the EH group). Parity and mean menopausal age, however, showed no significant differences between the groups. The EC group had a mean parity of 1.52 children, while the EH group had a mean parity of 1.62 children. The mean age of menopause in the EC group was 48.71 years, and 48.37 years in the EH group. Oral contraceptive use was more prevalent in the EC group (37.5%) compared with the EH group (25.5%). Our analysis revealed a difference in BMI between the two groups; patients in the EC group had a higher mean BMI (36.58 ± 5.598) than those in the EH group (29.945 ± 5.315). These data are detailed in Table 1.

Histopathological analysis of the EC specimens revealed a predominance of endometroid carcinomas (85.43%). Less frequent histological subtypes included mixed carcinomas (4.69%), serous carcinomas (4.17%), clear cell carcinomas (3.65%), mucinous carcinomas (2.08%), and carcinosarcomas (1.56%).

Furthermore, FIGO staging of the EC cases demonstrated that the majority (71.87%) presented at stage I. Stage III disease was also observed in a significant proportion of patients (18.22%). Stages 0, II, and IV were less prevalent (2.60%, 6.25%, and 1.04%, respectively).

Lymph node involvement assessment indicated that most cases (88.54%) were lymph node-negative. Conversely, a minority of cases (11.46%) exhibited positive lymph node involvement. These data are presented in Table 2.

Analysis of RBC indices revealed no significant differences between the EC and EH groups for erythrocyte count (mean difference = 0.048, *p* = 0.404), haemoglobin concentration (mean difference = 0.299, *p* = 0.090), haematocrit (mean difference = 0.779, *p* = 0.104), or RDW (mean difference = 0.216, *p* = 0.707).

No statistically significant differences were observed in WBCs between the groups (mean difference = −0.049, *p* = 0.833). Similarly, granulocyte count (mean difference = 0.133, *p* = 0.549) and monocyte count (mean difference = 0.027, *p* = 0.142) did not exhibit significant variations.

Of interest, the EC group presented with a significantly lower mean lymphocyte count compared with the EH group (mean difference = −0.265, *p* = 0.000). Conversely, the EC group had a significantly higher mean platelet count (mean difference = 19.997, *p* = 0.004).

The MPV was also significantly elevated in the EC group compared with the EH group (mean difference = 0.292, *p* = 0.000). However, no significant difference was detected in PDW between the groups (mean difference = 0.693, *p* = 0.000).

CRP levels were significantly elevated in the EC group compared with the EH group (mean difference = 30.784, *p* = 0.000). Likewise, the EC group displayed a significantly higher mean fibrinogen level than the EH group (mean difference = 114.071, *p* = 0.000).

The NLR and PLR were significantly higher in the EC group than in the second group (*p* = 0.004 and *p* = 0.000, respectively). The MLR was also considerably elevated in the EC group (mean difference = 3.012, *p* = 0.000). These data are presented in Table 3 and Table 4 and also in Figure 1, Figure 2 and Figure 3.

Data shown in Table 5 and in Figure 4 summarize the diagnostic performance of different markers in differentiating between the two groups (the EC group being noted as the positive group and, respectively, the EH group being noted as the negative group). CRP was observed to be the strongest predictor, with an area under the curve (AUC) of 0.961 (*p*-value < 0.000). Lymphocytes appeared to be the weakest predictor, with an AUC of 0.377.

Regarding our main topic of interest, ROC curve analysis was employed to assess the diagnostic efficacy of NLR and PLR for differentiating EC from EH. While both NLR and PLR demonstrated statistically significant discriminatory ability (*p*-value < 0.001), their AUC values indicated a moderate level of diagnostic accuracy. The AUC for NLR was 0.619 [standard error (SE) = 0.025, 95%, CI: 0.570–0.66]). PLR achieved a marginally higher AUC of 0.632 (SE = 0.024, 95% CI: 0.584–0.680). These findings suggest that NLR and PLR may have limited standalone utility in EC diagnosis due to their moderate discriminatory power.

In the following grouping of factors (see Figure 5 and Table 5), CRP continued to be the best marker for differentiation between groups, based on AUC (0.961), and lymphocytes remained the worst discriminator (AUC = 0.377). Several other markers showed moderate discriminatory performance (AUC between 0.5 and 0.7), including platelets, MPV, PDW, NLR, PLR and MLR.

When considering pairing of all serological biomarkers and selected baseline characteristics (see Figure 6 and Table 5), CRP was still the strongest marker to distinguish between groups, based on AUC (0.961), followed by fibrinogen (AUC = 0.857) and BMI (AUC = 0.818). In this scenario, menarche (AUC = 0.047) appeared to be the least discriminating marker, but we have to take into account that in the EC group the mean age was lower compared with the EH group.

## 4. Discussion

Investigating CBC indices, inflammatory markers, and the specified ratios between EC and EH groups yielded several noteworthy findings.

In the present study, there were no significant differences regarding RBC count, hemoglobin concentration, hematocrit, or RDW between the groups. This indicates that neither the EC diagnosis nor the EH condition substantially impacted RBC production or function. RBCs, which are the oxygen carriers, may display compensatory numerical and functional alterations, such as anaemia, due to the increased oxygen demand of the tumoral cells [14,15]. A Turkish study that included 416 patients divided into three groups (EC, EH, and healthy controls), through measuring the mean corpuscular volume (an indicator of average red blood cell size), observed a significant increase (*p*-values of 0.018 and 0.001, respectively) within both the EC and EH groups compared with the healthy control group. In contrast, RDW (a measure of the variation in RBC size) was found to have significantly lower values in patients with EC compared with the control and EH groups (*p* < 0.01), suggesting a more uniform RBC size distribution within the EC group compared with the others [16].

A Korean study investigating 431 patients with EC identified a preoperative median RDW of 12.8% (ranging from 11.1% to 27.8%). Patients were divided into groups with high (>12.8%) or low RDW (≤12.8%) based on this value. Statistically significant associations (*p* < 0.05) were observed between high RDW and several patient characteristics, including age, BMI, FIGO stage, pelvic lymph node metastases, and risk of recurrence. Furthermore, the prevalence of high RDW progressively increased across FIGO stages, with the highest proportion observed in stage IV. Specifically, the percentages of patients with high RDW were 14.4% in stage I, 32.2% in stage II, 37.6% in stage III, and 39.4% in stage IV. This suggests a potential role for RDW as a prognostic marker in EC [17].

Intriguingly, while total WBC count, granulocyte count, and monocyte count exhibited no statistically significant differences between the EC and EH groups, the lymphocyte count presented a distinct pattern in the present study. The EC group displayed a significantly lower lymphocyte count than the EH group. The association between elevated WBC and neoplasia was investigated in a prior prospective cohort study and subsequently linked in multivariate models to four types of cancer.

Specifically, women within the highest quartile of WBC count (6.80–15.00 × 10^9^ cells/L) exhibited a demonstrably increased hazard ratio (HR) for the following cancers compared with those in the lowest quartile (2.50–4.79 × 10^9^ cells/L): invasive breast cancer (HR 1.15, 95% CI 1.04–1.26), colorectal cancer (HR 1.19, 95% CI 1.00–1.41), EC in postmenopausal women (HR 1.42, 95% CI 1.12–1.79), and lung cancer (HR 1.63, 95% CI 1.35–1.97) [18]. A retrospective study originating from Turkey compared preoperative WBC count between 177 patients diagnosed with EC and 100 patients with benign gynecological conditions and noted the potential significance of preoperative WBC count above 10,500/mm^3^ as an independent biomarker for both the diagnosis and staging of EC (stages I–III vs. IV), with good sensitivity (88.9%) indicating its ability to identify a high proportion of true EC cases accurately. However, the positive predictive value (25.8%) was lower, implying that a high WBC count alone may not be sufficient for a definitive EC diagnosis and necessitating further evaluation [19].

A retrospective analysis was conducted in Korea to evaluate the potential of preoperative blood markers for diagnosing EC. The study included 238 women diagnosed with EC and 596 healthy controls and observed a significant increase (*p*-value < 0.001) in neutrophil, lymphocyte, and monocyte counts within the EC group, in contrast with eosinophil and basophil levels, which did not exhibit a significant difference between the groups (*p*-value = 0.64 and 0.523, respectively) [20].

A recent Chinese study investigated the association between immune cell levels and EC stage. The analysis involving 121 EC patients and 300 healthy controls revealed significantly lower CD4+ T lymphocyte percentages in the EC group (*p* = 0.013). CD4+ T cells are essential for coordinating the immune response, suggesting a potential link between weakened immunity and EC development [21]. Furthermore, a Korean study examined the presence of CD4+ and CD8+ T lymphocytes within endometrial endometrioid adenocarcinoma tissues. Compared to the control groups, tumor samples exhibited significantly higher proportions of both CD8+ (67.4%) and CD4+ (44.9%) T lymphocytes (*p* < 0.05). Notably, a negative correlation emerged between the extent of both CD4+ and CD8+ lymphocyte infiltration and histological grade and myometrial invasion depth [22].

Conversely, in our study, the EC group displayed a significantly higher platelet count and MPV, denoting increased platelet size, which was potentially indicative of an activated state. However, the absence of a significant difference in PDW suggests minimal variation in platelet size within the EC group. Platelets, essential for blood clotting, may manifest as thrombocytosis (>400 × 10^3^ platelets/mm^3^), which has been shown to be an independent predictor of OS regardless of FIGO stage and also predict poor disease-free survival (DFS) and progression-free survival (PFS) [23].

Also, a retrospective cross-sectional survey conducted in Turkey revealed no significant differences between EC, EH, and healthy controls regarding platelet count (PC). However, platelet indices indicative of size and distribution—MPV, PDW, and plateletocrit (PCT)—were all significantly elevated (*p* < 0.001) in the EC group compared with the control group [11]. A systematic review to evaluate the association between MPV and EC identified eight studies encompassing 1707 patients from China and Turkey. Notably, all included studies that incorporated a control group consistently reported a significant increase in MPV levels within the EC patient population compared with healthy controls [24].

In our study, the EC group presented with significantly higher NLR, PLR, and MLR compared with the EH group.

Consistent with current observations, previous studies have established independent correlations between NLR and PLR with tumor stage, grade, and OS in patients diagnosed with EC [25]. Notably, an elevated NLR has been associated with an increased risk of lymph node metastasis [26], and cervical stromal involvement [4]. It has been suggested that through applying a cut-off value of ≥13.50 mm for endometrial thickness (sensitivity 75%, specificity 83.6%) and ≥2.20 for NLR (sensitivity 81.3%, specificity 60.5%), the diagnostic accuracy for EC detection may improve [27]. Also, Korean research observed a significant elevation (*p*-value = 0.012 and *p*-value < 0.001, respectively) in two key WBC ratios: the NLR and the multiplication of neutrophils and monocytes (MNM) within the EC group compared with controls [20].

A meta-analysis encompassing 14 studies and 5274 patients was conducted in China to evaluate the prognostic significance of NLR, PLR, and MLR in EC. NLR and PLR demonstrated a significant association with OS in both univariate analysis (NLR: HR, 2.51; 95% CI, 1.70–3.71; *p* < 0.001; PLR: HR, 2.50; 95% CI, 1.82–3.43; *p* < 0.001) and multivariate analysis (NLR: HR, 1.87; 95% CI, 1.34–2.60; *p* < 0.001; PLR: HR, 1.86; 95% CI, 1.22–2.83; *p* = 0.004), as opposed to MLR (univariate analysis: HR, 1.44; 95% CI, 0.70–2.95; *p* = 0.325; multivariate analysis: HR, 1.01; 95% CI, 0.39–2.60; *p* = 0.987). Furthermore, NLR and PLR were significantly associated with DFS in the univariate analysis (NLR: HR, 2.50; 95% CI, 1.38–4.56; *p* = 0.003; PLR: HR, 1.91; 95% CI, 1.30–2.81; *p* = 0.001), NLR remaining significant in the multivariate analysis (HR, 2.06; 95% CI, 1.26–3.37; *p* = 0.004), as opposed to MLR, which lacked a significant association with DFS (UA: HR, 0.36; 95% CI, 0.03–4.13; *p* = 0.409) [28].

Likewise, Chinese research evaluated the distribution of NLR, PLR, and MLR in a cohort consisting of 1111 patients with EC with a median follow-up of 40 months (median age: 56 years) and observed the association between NLR, PLR, MLR, and OS. Median values (range) for these ratios were NLR: 2.01 (0.52–60.44); PLR: 121.11 (24.06–634.48); MLR: 0.19 (0.01–0.83). Following the determination of optimal cut-offs (NLR: >2.14, PLR: >131.82, MLR: >0.22), patients were stratified for subsequent analysis. Multivariate analysis revealed a statistically significant association between elevated levels of each marker and poorer OS: high NLR (>2.14): HR 2.71, 95% CI: 1.83–4.02, *p* < 0.001; high PLR (>131.82): HR = 2.75 (95% CI: 1.90–3.97), *p* < 0.001; high MLR (>0.22): HR = 1.72 (95% CI: 1.20–2.45), *p* = 0.003. Interestingly, a combined indicator encompassing all three elevated markers (high NLR + high PLR + high MLR) demonstrated the most robust prognostic value (HR = 4.34, 95% CI: 2.54–7.42, *p* < 0.001) [12].

In our study, the EC group presented significantly higher CRP levels compared with the EH group.

Several studies support a link between inflammatory markers and EC. A case–cohort study highlighted positive correlations between pre-diagnostic blood levels of CRP, IL-6, and TNF-α with EC incidence. Interestingly, only CRP remained significantly associated with EC risk after adjusting for BMI, suggesting a potentially independent role for CRP in disease development [29]. Similar findings were reported for Canadian research that included 549 patients with histologically confirmed EC, identifying CRP, but not IL-6 or TNF-α, as a risk factor for type I EC. Additionally, that study identified a statistically significant interaction between BMI and CRP levels in influencing EC risk. Specifically, the strength of the association between CRP and EC progressively increased with higher BMI values. Those findings suggest that elevated CRP may not function as an independent risk factor for EC but rather acts as a potential moderator, amplifying EC risk in the presence of obesity, particularly central obesity [30].

Furthermore, a retrospective Japanese study that reviewed CRP levels and CRP-to-albumin ratio defining the Glasgow Prognostic Score (GPS) in 431 patients with EC noted a significant association between high GPS (GPS 2) and poorer clinical outcomes. Patients with GPS 2 exhibited demonstrably shorter PFS and OS compared with those with a lower GPS (0 + 1) (*p* < 0.001 for both PFS and OS). Moreover, multivariate analysis independently validated GPS 2 as a robust predictor of both disease recurrence (*p* < 0.001) and patient mortality (*p* < 0.001) across the entire EC cohort. In the context of GPS, incorporating albumin concentrations alongside CRP levels may enhance the prognostic accuracy compared to solely serum CRP [31].

Also, according to the present study, the EC group presented significantly higher fibrinogen levels than the EH group.

Chinese research that included 942 patients with EC identified a statistically significant correlation (*p*-value = 0.049) between elevated plasma fibrinogen levels and OS in patients with EC. In that study, preoperative plasma fibrinogen concentration exceeding 3.25 g/L demonstrated a markedly increased HR of 1.807 contributing to poorer OS compared with those with lower fibrinogen levels (95% CI: 1.003–3.253). Significant variations in fibrinogen levels were noted across patient subgroups defined by age, menopausal status, BMI, FIGO stage, tumor grade, histological type, myometrial invasion depth, presence of LVSI, and comorbidities [32]. Likewise, according to an Austrian multi-institutional retrospective study, plasma fibrinogen concentration can be used as an independent prognostic parameter for DFS and OS in EC patients [33].

## 5. Conclusions

In summary, this analysis revealed several significant differences in serological biomarkers (lymphocytes, platelets, MPV, NLR, PLR, MLR, CRP, fibrinogen levels) tested between patients with EC and EH, suggesting potential alterations in the immune system and hemostatic function in EC-diagnosed patients. Our results demonstrate the existence of biomarkers that are easily obtainable and affordable for any health system, regardless of the country’s level of development, for a globally prevalent disease. While individual markers, such as NLR and PLR, may offer limited standalone diagnostic utility due to their moderate discriminatory power, their potential lies in the development of a multi-marker panel. Integrating these biomarkers with established clinical data, including patient history and imaging, holds promise for significantly enhancing diagnostic accuracy for EC. Further research with larger patient cohorts is warranted to refine marker-specific cut-off points and optimize their discriminatory power. This will facilitate the translation of these findings into a cost-effective and widely applicable diagnostic tool.

## Figures and Tables

**Figure 1 cancers-16-01935-f001:**
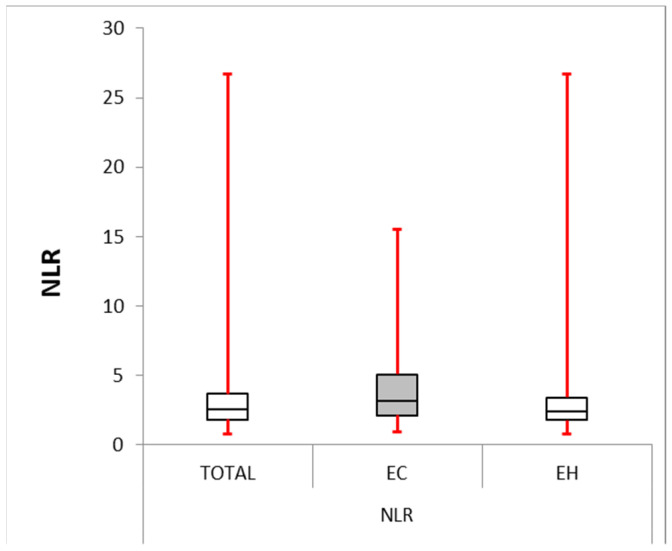
Neutrophile–lymphocyte ratio: box plot comparing NLR between the EC and EH groups. The median NLR was significantly higher in the EC group (4.191) compared with the EH group (2.396, *p*-value = 0.004). The interquartile range (IQR) for the EC group (2.128–5.063) was larger than for the EH group (1.786–3.421), suggesting greater variability in NLR within the EC group.

**Figure 2 cancers-16-01935-f002:**
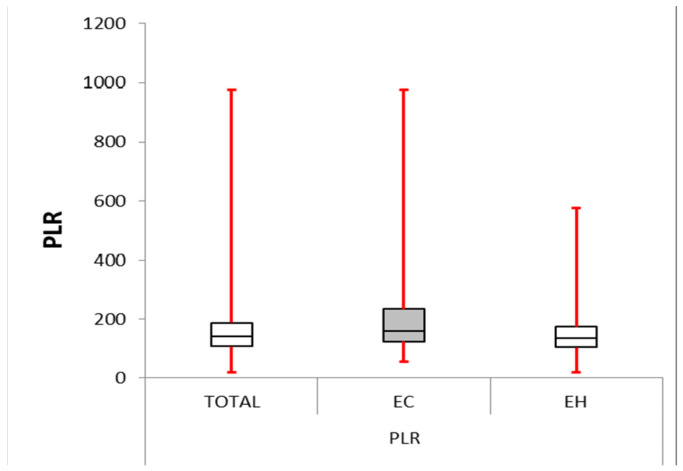
Platelet–lymphocyte ratio: box plot comparing PLR between the EC and EH groups. The median PLR was significantly higher in the EC group (161.204) compared with the other group (136.878, *p*-value < 0.0001). The IQR for the EC group (123.986–234.329) was more significant than for the EH group (105.089–174.808), suggesting greater variability in PLR within the EC group.

**Figure 3 cancers-16-01935-f003:**
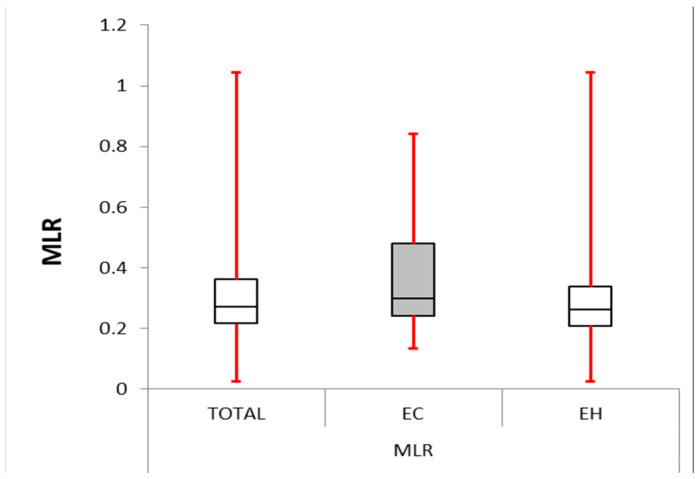
Monocyte–lymphocyte ratio: box plot comparing MLR between the EC and EH groups. The median MLR was significantly higher in the EC group (0.298) compared with the other group (0.262, *p*-value < 0.0001). The IQR for EC group (0.243–0.480) was larger than the control group (0.208–0.337), suggesting greater variability in MLR within the EC group.

**Figure 4 cancers-16-01935-f004:**
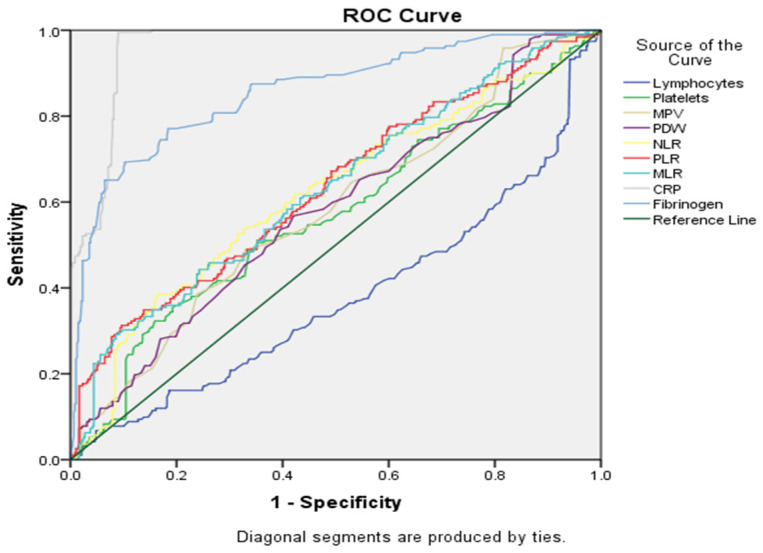
ROC curve for statistically significant markers.

**Figure 5 cancers-16-01935-f005:**
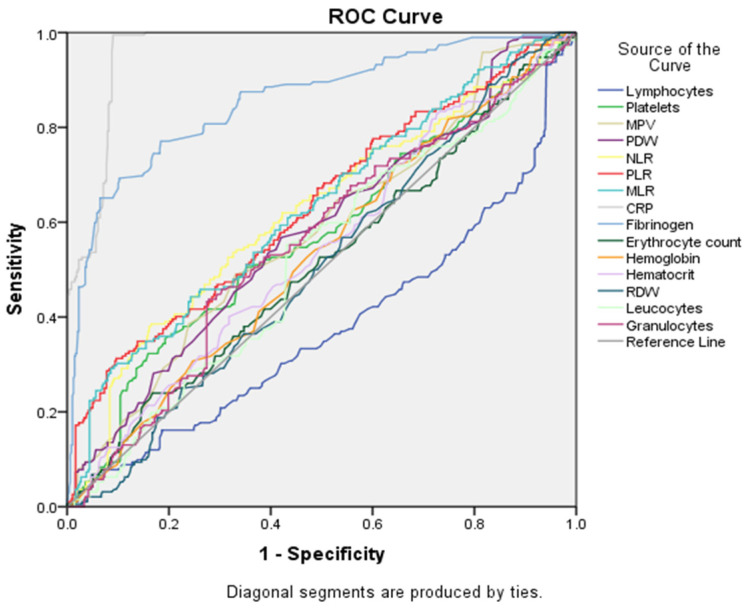
ROC curve analysis for all serological markers tested.

**Figure 6 cancers-16-01935-f006:**
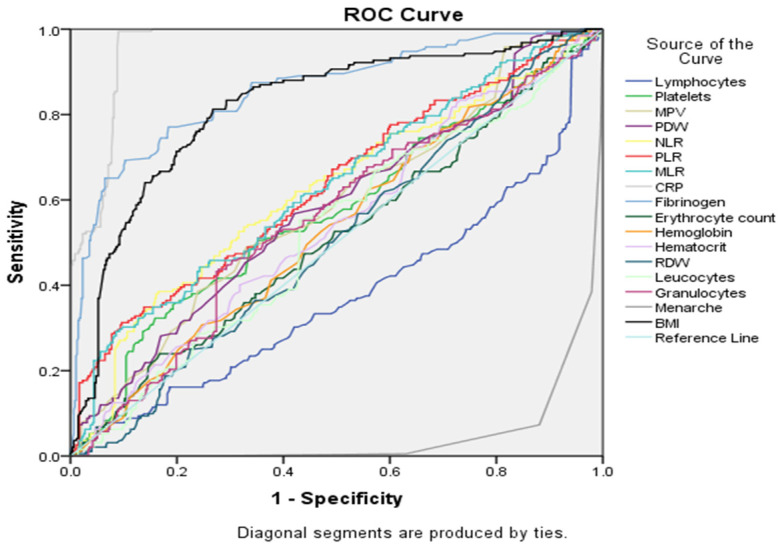
ROC curve analysis for all serological markers and selected baseline characteristics.

**Table 1 cancers-16-01935-t001:** Baseline clinical characteristics: Descriptive data: (a) EC group; (b) EH group.

**Statistic (EC)**	**Age (y)**	**Menarche (y)**	**Parity**	**Menopause (y)**	**Weight (kg)**	**Height (m)**	**BMI (kg/m^2^)**
Nbr. of observations	192	192	192	192	192	192	192
Minimum	33.000	8.000	0.000	30.000	60.000	1.470	21.971
Maximum	93.000	14.000	5.000	59.000	167.000	1.780	55.799
1st Quartile	55.750	11.000	1.000	47.000	87.000	1.580	33.100
Median	63.000	11.000	1.000	50.000	95.000	1.600	37.254
3rd Quartile	69.250	12.000	2.000	52.000	99.250	1.640	39.257
Mean	62.417	11.109	1.521	48.708	94.724	1.611	36.583
Variance (n − 1)	112.69	1.218	1.288	28.229	208.724	0.002	31.335
SD (n − 1)	10.616	1.104	1.135	5.313	14.447	0.049	5.598
**Statistic (EH)**	**Age (y)**	**Menarche (y)**	**Parity**	**Menopause (y)**	**Weight (kg)**	**Height (m)**	**BMI (kg/m^2^)**
Nbr. of observations	478	478	478	478	478	478	478
Minimum	25.000	9.000	0.000	25.000	50.000	1.450	20.077
Maximum	87.000	18.000	9.000	57.000	170.000	1.810	58.824
1st Quartile	46.000	13.000	1.000	45.000	71.250	1.600	26.624
Median	50.000	14.000	2.000	50.000	78.000	1.640	29.044
3rd Quartile	59.750	14.000	2.000	52.000	87.000	1.680	32.667
Mean	53.119	13.770	1.615	48.367	80.399	1.640	29.945
Variance (n−1)	116.55	1.255	1.323	23.912	198.638	0.004	28.254
SD (n−1)	10.796	1.120	1.150	4.890	14.094	0.062	5.315

**Table 2 cancers-16-01935-t002:** Histopathological characteristics of EC patients (Descriptive data).

**Histological Type of EC**	**No (%)**
Endometrioid Adenocarcinoma	164 (85.42)
Villoglandular variant	76 (39.58)
Variant with squamous differentiation	66 (34.38)
Cillated cell variant	22 (11.46)
Mixed carcinoma	9 (4.69)
Endometrioid and serous	2 (1.04)
Serous and clear cell	2 (1.04)
Endometrioid and mucinous	1 (0.52)
Endometrioid and clear cell	1 (0.52)
Serous carcinoma	8 (4.17)
Clear cell carcinoma	7 (3.65)
Mucinous carcinoma	4 (2.08)
Carcinosarcoma	3 (1.56)
**Lymph node status**	**No (%)**
Negative	170 (88.54%)
Positive	22 (11.46%)
**LVSI**	
Negative	166 (86.46%)
Positive	26 (13.54%)
**Grade**	No (%)
G1	84 (43.75%)
G2	65 (33.85%)
G3	43 (22.40%)
**FIGO stage**	No (%)
FIGO 0	5 (2.60%)
FIGO IA	62 (32.29%)
FIGO IB	76 (39.58%)
FIGO II	12 (6.25%)
FIGO IIIA	5 (2.60%)
FIGO IIIB	14 (7.29%)
FIGO IIIC1	12 (6.25%)
FIGO IIIC2	4 (2.08%)
FIGO IVB	2 (1.04%)
**Total**	**192 (100.00%)**

**Table 3 cancers-16-01935-t003:** Comparison between hematological markers obtained from blood samples of patients diagnosed either with EC or EH. Summary statistics (quantitative data): EC vs. EH.

			*t*-Test for Equality of Means					
	EC (Group 1)	EH (Group 2)					95% Confidence Interval (CI) of the Difference	
	Min–Max (Median) Mean ± SD	Min–Max (Median) Mean ± SD	t	df	Sig. (2-Tailed)	Mean Difference	Lower	Upper
RBC count (×10^6^/μL)	2.05–11.71 (4.47) 4.46 ± 0.80	1.96–6.71 (4.45) 4.41 ± 0.61	0.834	668	0.404	0.048	−0.065	0.1620
Haemoglobin (gr/dL)	6.0–16.1 (13.0) 12.65 ± 1.83	3.7–16.3 (12.8) 12.35 ± 2.17	1.68	668	0.09	0.299	−0.050	0.6490
Haematocrit (%)	18.5–47.1 (39.2) 38.40 ± 5.16	14.2–48.9 (38.7) 37.62 ± 5.72	1.629	668	0.104	0.779	−0.160	1.7180
RDW (fL)	33.2–66.4 (42.6) 43.26 ± 4.12	12.5–79.1 (42.5) 43.05 ± 7.49	0.377	668	0.707	0.216	−0.909	1.340
WBCs (×10^3^/μL)	2.59–17.26 (8.22) 8.34 ± 2.40	3.32–25.81 (7.775) 8.39 ± 2.850	−0.211	668	0.833	−0.0493	−0.50736	0.4087
Granulocytes (×10^3^/μL)	1.66–12.80 (5.90) 5.89 ± 2.10	1.38–26.27 (5.15) 5.76 ± 2.75	0.600	668	0.549	0.1327	−0.3017	0.5671
Lymphocytes (×10^3^/μL)	0.18–4.31 (1.69) 1.86 ± 0.72	0.460–10.57 (2.01) 2.12 ± 0.79	−4.014	668	0.000	−0.2648	−0.3944	−0.1353
Monocytes (×10^3^/μL)	0.03–1.22 (0.59) 0.60 ± 0.19	0.04–1.77 (0.55) 0.57 ± 0.22	1.471	668	0.142	0.0272	−0.0091	0.0635
Platelets (×10^3^/mm^3^)	132–655 (307) 304.4 ± 84.24	45–571 (277.5) 284.42 ± 78.65	2.915	668	0.004	19.997	6.527	33.467
MPV (fL)	8.20–13.5 (10.8) 10.82 ± 1.02	7.80–13.80 (10.5) 10.53 ± 0.98	3.432	668	0.000	0.292	0.125	0.460
PDW (fL)	8.80–19.80 (12.80) 13.14 ± 2.34	0.60–21.0 (12.20) 12.44 ± 2.25	3.554	668	0.000	0.693	0.310	1.076

**Table 4 cancers-16-01935-t004:** Comparison between hematological ratios and inflammatory markers obtained from blood samples of patients diagnosed either with EC or EH. Summary statistics (quantitative data): EC vs. EH.

			*t*-Test for Equality of Means					
	EC (Group 1)	EH (Group 2)					95% CI of the Difference	
	Min–Max (Median) Mean ± SD	Min–Max (Median) Mean ± SD	t	df	Sig. (2-Tailed)	Mean Difference	Lower	Upper
NLR	0.96–15.55 (4.19) 3.52 ± 1.97	0.77–26.69 (2.39) 2.96 ± 2.35	2.893	668	0.004	0.5573	0.1790	0.9356
PLR	54.77–977.77 (161.20) 189.90 ± 102.25	20.15–574.07 (136.87) 147.68 ± 63.66	6.441	668	0.000	42.2155	29.3465	55.0845
MLR	0.13–0.84 (0.29) 0.36 ± 0.16	0.02–1.04 (0.26) 0.29 ± 0.13	5.37	668	0.000	0.07	0.045	0.094
CRP (mg/L)	0.50–270.40 (10.92) 31.91 ± 38.78	0.04–88.80 (0.29) 1.13 ± 4.87	17.039	668	0.000	30.7840	27.2365	34.3315
Fibrinogen (mg/dL)	220–664 (420) 429.08 ± 86.78	110–632 (315) 315.01 ± 65.57	18.471	668	0.000	114.071	101.945	126.197

**Table 5 cancers-16-01935-t005:** Area Under the Curve for all serological markers and selected baseline characteristics.

Test Result Variable(s)	Area	Std. Error ^a^	Asymptotic Sig. ^b^	Asymptotic 95% Confidence Interval	
				Lower Bound	Upper Bound
Lymphocytes	0.377	0.025	0.000	0.328	0.426
Platelets	0.571	0.025	0.004	0.521	0.620
MPV	0.582	0.024	0.001	0.534	0.630
PDW	0.579	0.024	0.001	0.531	0.627
NLR	0.619	0.025	0.000	0.570	0.667
PLR	0.632	0.024	0.000	0.584	0.680
MLR	0.629	0.024	0.000	0.582	0.677
CRP	0.961	0.007	0.000	0.949	0.974
Fibrinogen	0.857	0.017	0.000	0.824	0.891
Erythrocyte count	0.512	0.025	0.627	0.463	0.561
Hemoglobin	0.534	0.024	0.166	0.487	0.582
Hematocrit	0.541	0.025	0.096	0.493	0.589
RDW	0.507	0.024	0.773	0.460	0.554
Leucocytes	0.521	0.025	0.398	0.473	0.569
Granulocytes	0.555	0.025	0.025	0.507	0.603
Menarche	0.047	0.008	0.000	0.032	0.062
BMI	0.818	0.019	0.000	0.781	0.854

The test result variable(s): lymphocytes, platelets, MPV, PDW, NLR, PLR, MLR, CRP, fibrinogen, erythrocyte count, hemoglobin, hematocrit, RDW, leucocytes, granulocytes, menarche, and BMI all have at least one tie between the positive actual state group and the negative actual state group. Statistics may be biased. ^a^. Under the nonparametric assumption ^b^. Null hypothesis: true area = 0.5.

## Data Availability

All data are available in the external and internal archive (database) of the Emergency Clinical Hospital of Elias, Bucharest, Romania.

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
