# Peer review of "Clinical Correlations between Serological Markers and Endometrial Cancer"

_cancers, 2024, doi:10.3390/cancers16101935_

Round 1

Reviewer 1 Report

Comments and Suggestions for Authors

This manuscript describes a retrospective study that addresses the question of whether standard blood markers can reliably distinguish endometrial neoplasia from endometrial hyperplasia. Using various blood markers, several studies reported in the literature suggest that such differences may be valid, and the authors of the present study have extended this approach. It remains to be seen, however, if such measurements will prove sufficiently reliable to be used in countries with marginal health care systems.

The authors provide a thorough description of the patient background, including the histopathological characteristics of those with endometrial cancer, the methodology used, and their results. Several issues have arisen that need to be addressed before publication can be recommended.

1.     Table 3 and interpretation. Not surprisingly for human clinical samples, the standard deviations tend to be quite large (cf. Table 3). With this in mind, it is indeed surprising that statistical significance was obtained with lymphocytes, platelets, MPV, NLR, PLR, MLR, CRP, and fibrinogen. Focusing, for example, on CRP that exhibits the greatest difference in means values between endometrial cancer (31.918) and endometrial hyperplasia (1.134), the standard deviations are, respectively, 38.787 and 4.875. It is questionable how significance can be achieved with such a large standard deviation in the cancer group. The closeness of the means, coupled with the relatively large standard deviations of the other markers claimed to show significant differences, raises questions on how measurement of these markers on a given patient could yield reliable diagnostic information. Another interpretation of these results is that such measurements are not yet to a point of being applicable in a clinical setting.

2.     Table 3: (a) It would benefit the reader if a space or line separated each of the blood markers. (b) In the heading, Sig, with (2-tailed) below it, the closing parenthesis, ), should be moved up to give (2-tailed) on one line. (c) For the markers, RBC count, haemoglobin, and haematocrit, minima and maxima are given to one or two significant decimal places, yet the median is presented to three decimal places for RBC count and for the mean ± SD for the three blood markers. Can the authors truly justify the added significance? (d) It is taxing to the reader to have each of Tables 1-3 appear on two pages; however, it would require major reformatting by the layout editor to overcome this issue.

3.     The ordinates on panels a and b of Figures 1-3 are shown with different scales. This can be confusing to the reader; why not plot the graphs using the same scale?

4.     Typographical corrections: change MRL to MLR on line108 and in Table 3.

5.     It would be helpful to the reader if some of the long paragraphs in the Discussion could be broken into shorter paragraphs. For example, new paragraphs could begin on lines 282 and  329.

Reviewer 2 Report

Comments and Suggestions for Authors

Line 97: The abbreviations NLR, PLR, MLR appear here for the first time but are not explained until lines 107-109. Please move the explanations to line 97 where abbreviations are mentioned.

Line 177: The average menarche in EH group is 12.77 year here but is 13.77 in table 1. Please correct. Furthermore, if the average menarche is statistically different between EH and EC groups, this fact should be mentioned as you did for BMI.

For Figure 1-3, the subplots (a) and (b) are redundant. Please only show (c). The y-label of (c) is missing and the title of boxplots in (c) is unnecessary. "XLSTAT Trial" in (c) should be removed properly to reach the publication quality. The group names in (c) are inconsistent from Figure 1 to 3. 

The legend in Figure 4 is too small and in poor resolution.

The authors only tried ROC curve analysis for single variables such as NLR or PLR, but it is not surprising that single variable has very limited diagnostic power. The authors should also try ROC curve analysis based on the combination of multiple variables. There are many binary classification tools available for multi-variable data, such as support vector machine and perceptron. Please perform such a multiple variable analysis in three scenarios: (1) including selected haematological markers (e.g., all variables in table 3 with statistical significance); (2) including all haematological markers; (3) including all haematological markers plus selected baseline clinical characteristics (such as BMI and menarche).

Round 2

Reviewer 1 Report

Comments and Suggestions for Authors

No comments or further suggestions.

Reviewer 2 Report

Comments and Suggestions for Authors

N/A.